# Seasonal Variation of the Mobility and Toxicity of Metals in Beijing's Municipal Solid Waste Incineration Fly Ash

**Hang Zhao [1], Yang Tian [1], Rong Wang [1], Rui Wang [1], Xiangfei Zeng [1], Feihua Yang [2], Zhaojia Wang [2], Mengjun Chen [1],* and Jiancheng Shu [1]**

[1] Key Laboratory of Solid Waste Treatment and Resource Recycle, Ministry of Education, Southwest University of Science and Technology, Mianyang 621010, China; zhaohang1016@163.com (H.Z.); tianyang941005@gmail.com (Y.T.); wr276432218@163.com (R.W.); wangrui6393@163.com (R.W.); zxfforeverup@163.com (X.Z.); shujc@swust.edu.cn (J.S.)

[2] State Key Laboratory of Solid Waste Reuse for Building Materials, Beijing Building Materials Academy of Sciences Research, Beijing 100041, China; chyangfeihua@126.com (F.Y.); zhaojiaw@bbma.com.cn (Z.W.)

* Correspondence: kyling@swust.edu.cn

**Abstract:** Metal mobility and toxicity of the municipal solid waste incineration (MSWI) fly ash from different seasons in Beijing were studied using leaching toxicity procedures, sequential extraction procedures, and bioavailability procedures. The X-ray diffraction results showed that MSWI fly ash contained $CaSO_4$, $CaCO_3$, and KCl. The Pb, Zn, and Cd contents of MSWI fly ash were high, especially during autumn, being 42, 77, and 1260 times higher than that of their soil backgrounds, respectively. Leaching toxicity experiments showed that MSWI fly ash is hazardous, since Pb exceeded the maximum threshold (5 mg/L). The main alkali metal ions and anions, heavy metals total content, leaching concentration, chemical speciation, and bioavailability varied seasonally. The Pb and Zn leaching concentrations in summer and autumn were higher than that of the other two seasons. Sequential extraction procedures indicated that Pb, Zn, and Cd showed extremely high metal mobility, i.e., the residual states of Pb and Cd in spring were 5.83% and 1.21%, respectively, and that of Zn in autumn was 10.68%. These results will help industries, governments, and the public better understand the risk of MSWI fly ash and will urge them to pay more attention to preventing harm to the ecosystem and human beings.

**Keywords:** MSWI fly ash; heavy metal; mobility; leaching toxicity; BCR

## 1. Introduction

Incineration has been applied for the disposal of municipal solid waste (MSW) for more than 100 years [1]. Currently, 130 million tons of MSW are burned globally every year. About 4.1 million tons of MSW are incinerated each year in China, leading to a large quantity of municipal solid waste incineration (MSWI) fly ash [2]. MSWI fly ash contains high levels of heavy metals (HMs) such as Zn, Pb, Cu, Cd, Cr, and Ni [3–5]. Their content and leaching concentration are regulated by a set of standards [6,7]. At present, separation/extraction [7], solidification/stabilization [4], and heat treatment [5,6] are used to reduce the harm of solid waste caused by HMs. However, heavy metal ions are release secondarily after treatment [4,7]. The released heavy metals may transfer to ecosystems such as groundwater or crops through water and food webs, thus affecting human health [8].

With the development of the study on the toxicity of HMs of MSWI fly ashes, the evaluation of harmful effects of HMs were assessed by the total concentration, mobility and leaching toxicity [9,10] instead of total concentration gradually [11,12]. This is of great significance for the evaluation and control of HM pollution. Research has confirmed that the strong mobility of HMs are generally caused by their soluble and exchangeable states [13]. The leaching risk of HMs from MSWI fly ash is also affected by the characteristics of fly ash

and the environment, such as weather and aging, leaching method, leaching time, pH, and liquid-to-solid ratio (L/S) [14,15]. Among them, pH is the main factor that affects the HMs leaching rate [16]. Increased relative amounts of sulfur and nitrogen oxide emissions can lead to acid rain, resulting in a lower pH in the environment and easier leaching of toxic heavy metals [17]. The decrease in pH and the increase in soluble chlorine in MSWI fly ash solutions increase the toxic leaching potential of HMs [18]. Maklawe's research shows that seasonal variation is one of the key factors affecting the generation and composition of MSW [19,20]. Therefore, the species and contents of HMs in MSWI fly ash should be seasonal. However, there is limited information about seasonal variations in the mobility and toxicity of HMs in MSWI fly ash.

In this paper, HMs migration and leaching toxicity of MSWI fly ash from Beijing were studied over four seasons. The basic physicochemical properties of MSWI fly ash were analyzed by X-ray fluorescence spectrum (XRF), X-ray diffraction (XRD), and scanning electron microscopy (SEM). The leaching toxicity of HMs was tested using the sulfuric acid and nitric acid method (HJ/T 299–2007) and the toxicity characteristic leaching procedure (TCLP) [18]. The Community Bureau of Reference (BCR) method was used to determine the speciation of HMs in MSMI fly ash [21]. In addition, a bioavailability extraction method was used to determine the bioavailability. The data and conclusions are helpful in evaluating the impact of MSWI fly ash on the environment in different seasons. Meanwhile, our results may provide a scientific basis for the prevention and treatment of HMs in MSWI fly ash.

## 2. Materials and Methods

### 2.1. Materials

MSWI fly ash samples were collected from the Lujiashan MSW plant in Beijing. The plant uses a mechanical grate process, and it mainly deals with the MSW from the main urban areas of Beijing and the surrounding counties, with a capacity of 750 t/d. Samples were collected every month for a year, and 5 kg of fly ash was collected on a fixed day every month. The samples were collected at the ash outlet of the flue gas purification device in the waste incineration plant. All MSWI fly ash samples were sieved through a 100-mesh sieve. Then, they were fully mixed using the mill ball and dried in an oven at 105 °C to a constant weight. In this study, all of the chemical reagents ($HNO_3$, $HCl$, $HF$, $H_2SO_4$, $CH_3COOH$, $NaOH$, $NH_4SCN$, $NH_4Fe(SO_4)_2 \cdot 12H_2O$, $AgNO_3$, $C_{15}H_{15}N_3O_2$, $NH_3 \cdot H_2O$, $CH_3COONH_4$, $HONH_2HCl$, $H_2O_2$, and EDTA–2Na) were of analytical grade. Deionized water was obtained after treatment by the water purification system (Advantage A10, Millipore, Burlington, MA, USA). The experiments were performed at room temperature.

The composition of the sample was analyzed using XRF (XRF, UltimaIV, Rigaku, Tokyo, Japan). The mineral phase of the sample was analyzed using an XRD (XRD, Axios MAX, PANalytical B.V, Almelo, Netherlands). The XRD investigations were conducted by a D/MAX 2500 using Cu Kα radiation. The scanning speed was 1°/min, and the range was 2θ = 5–85°. Jade 6.0 was used for analysis of the possible presence of substances [22]. The size and morphology of the samples were observed by SEM (SEM, FESEM, S-3400N, HITACHI, Tokyo, Japan) at 5–25 kV voltage after sputtering and gold spraying [23]. Thermogravimetric analysis (TG) was performed on a TA Q500 apparatus between 20 and 1000 °C with a heating rate of 10 K/min under $N_2$ atmosphere [24].

### 2.2. Element Analysis

According to "The Technical Specification for Soil Environmental Monitoring" (HJ/T 166–2004) [25], 0.20 g of dried MSWI fly ash, 6 mL of $HNO_3$ (69%), 2 mL of HCl (72%), and 2 mL of HF (28%) were added to a digestion tube. The digestion tube was then inserted into the Multiwave Pro and subjected to digestion at 180 degrees for 1 h (Anton Paar, Graz, Austria). After that, the liquid was diluted to 50 mL. HMs (such as Zn, Pb, Cu, Cd, Cr, Ni, and As) were determined by inductively coupled plasma–optical emission spectroscopy

(ICP–OES, Optima 8300, PerkinElmer, Waltham, MA, USA). Refer to the Supplementary Materials for quality control/quality assurance (QC/QA).

$Cl^-$: referring to the "Methods for Chemical Analysis of cement" (GB/T 176-2017) [26], 2.00 g of the ($m_{1-1}$) sample, 50 mL of $H_2O$, and of 50 mL $HNO_3$ were mixed and boiled for 1–2 min and 5 mL of $AgNO_3$ was further added and boiled, followed by filtering and dilution to 200 mL. Then, 5 mL of $NH_4Fe(SO_4)_2 \cdot 12H_2O$ was added to the filtrate and titrated with $NH_4SCN$ until a reddish-brown precipitate appeared. The consumption was recorded as $V_{1-1}$. In addition, the blank experiment was carried out according to the same steps above, and the volume of $NH_4SCN$ consumed was recorded as $V_{1-2}$. $W_{Cl^-}$ was determined by following equation.

$$\omega_{Cl^-} = 0.8865 \times \frac{v_{1-2} - v_{1-1}}{v_{1-2} \times m_{1-1}} \tag{1}$$

$K_2O$ and $Na_2O$: referring to the "Methods for Chemical Analysis of cement" (GB/T 176-2017) [26], 0.50 g of the ($m_{2-1}$) sample, 5 mL of HF, and 10 mL of $H_2SO_4$ were mixed until white smoke no longer appeared upon heating. Deionized water, methyl red, $NH_3 \cdot H_2O$, and $(NH_4)_2CO_3$ were then added to the crucible. Finally, the solution was filtered and HCl was added until the solution turned red. The contents of $K_2O$ and $Na_2O$ in the solution were determined by flame photometry, which were ($m_{2-3}$) and ($m_{2-3}$), respectively. The calculation formula was as follows.

$$\omega_{K_2O} = \frac{m_{2-2} \times 0.1}{m_{2-1}} \tag{2}$$

$$\omega_{Na_2O} = \frac{m_{2-3} \times 0.1}{m_{2-1}} \tag{3}$$

$SO_3$: according to the "Methods for Chemical Analysis of cement" (GB/T 176-2017) [26], 0.50 g of the ($m_{3-1}$) sample, 40 mL of $H_2O$, and 10 mL of HCl were mixed, then boiled for 8 min, and diluted to 250 mL; then, 10 mL of $BaCl_2$ was added and the solution was filtered for 18 h. Finally, the solid was burned for 45 min and weighed to obtain ($m_{3-2}$). In addition, the blank experiment was carried out according to the same steps above; the weight of solid obtained was $m_{03-2}$. The specific calculation formula was as follows:

$$\omega_{SO_3} = \frac{(m_{3-2} - m_{03-2}) \times 0.343}{m_{3-1}} \times 100 \tag{4}$$

*2.3. Leaching Toxicity*

2.3.1. Leaching Toxicity Using the Sulfuric Acid and Nitric Acid Method

According to the "Solid Waste–Extraction procedure for leaching toxicity–Sulfuric acid and nitric acid method" (HJ/T299–2007) [27], the leaching agent was prepared as follows: the mixture of $HNO_3$ (65%) and $H_2SO_4$ (98%) (AR) with mass ratio of 1:2 was added to 1 L of deionized water to adjust the pH value to $3.20 \pm 0.05$. For the leaching procedure, 10.00 g of the sample was mixed with the leaching agent at an L/S ratio of 10:1, and the mixture was oscillated for $18 \pm 2$ h ($30 \pm 2$ r/min). The mixture was then filtered using a 0.45 μm microporous membrane. Finally, the HMs in the samples were measured by ICP–OES. Details regarding the QC/QA are provided in the Supplementary Materials.

2.3.2. Toxic Leaching of Solid Waste: TCLP

As the pH of the sample dissolved in water exceeded 5.0, extractant 2 was chosen for the leaching test (for extractant 2, put 1.5 L of ultrapure water into a 2 L beaker and add $CH_3COOH$ to make the pH value $2.88 \pm 0.05$). Then, 75.00 g of the sample and extractant 2 were mixed with the L/S at 20:1 and was then oscillated for $18 \pm 2$ h ($30 \pm 2$ r/min) and filtered by a 0.45 μm microporous membrane. ICP–OES was used to measure the HMs. Details on the QC/QA assurance are provided in Supplementary Materials.

### 2.3.3. Leaching at Different Initial pH

The MSWI fly ash samples (4.00 g) were then mixed with 40 mL of ultrapure water in centrifuge tubes, and a pH adjusting solution was added to maintain the pH value of the liquid at 2, 4, 6, 8, 10, and 12 (pH adjusting solution: 1 part volume of nitric acid is mixed with 99 parts volume of water or 1 part volume of sodium hydroxide is mixed with 99 parts volume of water). Then, the solution was oscillated for $18 \pm 2$ h ($30 \pm 2$ r/min) and filtered using a 0.45 μm microporous membrane. ICP–OES was used to measure the HMs.

### 2.4. Chemical Speciation

HMs chemical speciation was carried out in accordance with the three steps outlined by the BCR (Table 1) [28]. All extractions were oscillated for 16 h and centrifuged for 20 min (4000 r/min). The supernatant was filtered using 0.45 μm microporous membrane and stored for testing. The residual was analyzed by the digest method. The HMs concentrations in the extracts and digested solutions were analyzed by ICP–OES. Each sample was measured three times and reported as an average.

**Table 1.** Detailed information of the modified four-step sequential extraction method.

| Target Phase | Extraction Agent and Conditions |
|---|---|
| F1: Acid–soluble fraction | 40 mL of 0.11 mol/L $CH_3COOH$; 16 h |
| F2: Reduction fraction | 40 mL of 0.5 mol/L $HONH_2HCl$; 16 h |
| F3: Oxidation fraction | 10 mL of 30% $H_2O_2$, dissolved at room temperature for 1 h; then, 10 mL of 30% $H_2O_2$ was added again and digested at 85 °C (water bath) for 1 h; and 50 mL of 0.5 mol/L $(NH_4)_2CO_3$ was added; pH = 2; 16 h |
| F4: Residual fraction | 6 mL of $HNO_3$, 2 mL of HCl, and 2 mL of HF; digested |

### 2.5. Metal Bioavailability

The sample (4.00 g) and EDTA–2Na solution (40 mL) were mixed together. The EDTA–2Na solution (purchased from Sinopharm Chemical Reagent Co. Ltd., Shanghai, China) was used to simulate the chelation and adsorption of HMS in MSWI fly ash by plant root or plant cell organic acid exudates [29]. NaOH was used to adjust the pH of the mixture to 7. Then, the solution was oscillated for 1 h and centrifuged to separate the liquid and solid. The HMs content was measured by ICP–OES. The QC/QA is provided in Supplementary Materials.

## 3. Results

### 3.1. Characteristics of MSWI Fly Ashes

The MSWI fly ash samples for each month were mixed evenly. The XRF results expressed in the form of oxides indicated that the main components of the fly ash were CaO (39.18%), Cl (18.38%), $SiO_2$ (10.02%), $SO_3$ (8.97%), and $Na_2O$ (5.99%). Some components made up less than 5%, such as $K_2O$ (4.47%), $Al_2O_3$ (3.81%), MgO (3.75%), and $Fe_2O_3$ (2.11%), and the rest comprised 3.32%. The high CaO content is caused by the excessive lime slurry that was injected to reduce the emission of acidic gases ($SO_2$ and HCl) [30]. The high chloride content mainly comes from the incineration of plastic and kitchen waste [24]. The high content of perchlorates and sulfates prevents MSWI fly ashes from being used directly in cement and concrete, which means that it must be pretreated before reuse [31].

As shown in Figure 1a, the main crystalline phases in MSWI fly ashes are KCl, NaCl, $CaSO_4$, $CaCO_3$, CaClOH, $Ca(OH)_2$, and $(Fe, Mg)_2Al_4Si_5O_{18}$. These results can be confirmed against the results from other research [32]. $CaSO_4$, NaCl, and KCl mainly came from the burning of fabric filter bags [33]. KCl and NaCl are easily volatilized and easily enter MSWI fly ashes during incineration since they have a low boiling point [34]. Anhydrite ($CaSO_4$), calcium hydroxy calcium chloride (CaCl(OH)), calcite $CaCO_3$, and portlandite $(Ca(OH)_2)$ are produced when lime is sprayed to remove acid gases [33]. $(Fe, Mg)_2Al_4Si_5O_{18}$ is a new crystalline phase in which Mg and Fe are mixed with aluminosilicates at a sintering temperature of 900 °C [35]. The crystalline phase of HMs could not be detected, which

may be due to the low content of HMs in MSWI fly ash or because they are presented amorphously and embedded in aluminosilicates or silicates [13].

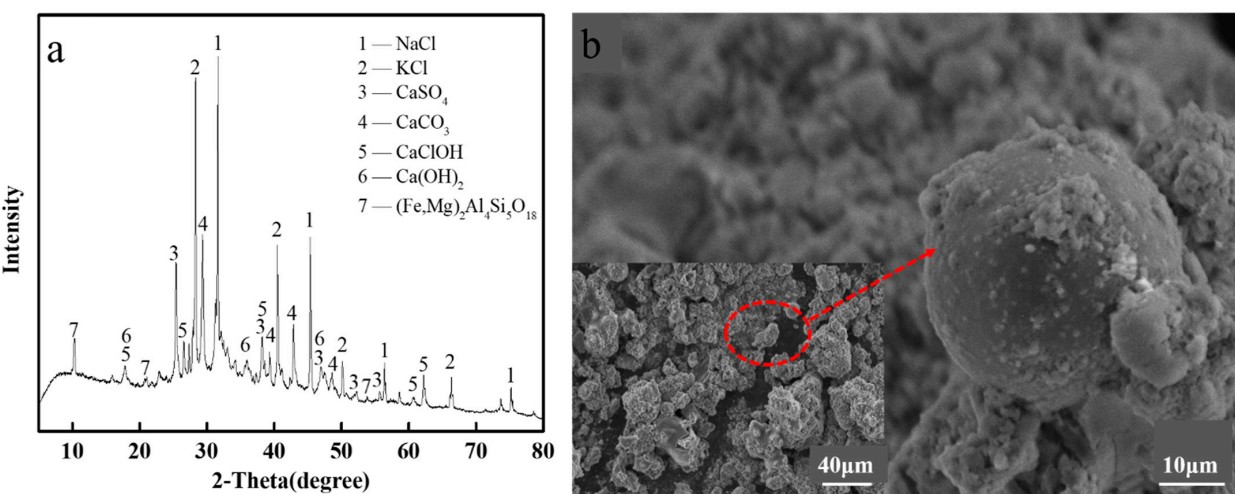

**Figure 1.** (**a**) XRD patterns of MSWI fly ash and (**b**) SEM images of MSWI fly ash.

As can be seen from Figure 1b, MSWI fly ashes are discrete and irregular. Furthermore, the irregular spherical magnification shows that the surface is rough. Many smaller particles are attached to the surface. The volatile HMs of MSWI fly ashes are easily leached due to the large surface particle spacing and high porosity, which can cause accumulation on the specific surface [36].

As shown in Figure 2, MSWI fly ashes show four obvious weight loss peaks. The first weight loss is 3.50% at 23–200 °C, and it is characterized by a weight loss peak at 100 °C. According to previous studies, this weight loss near 100 °C is caused by the evaporation of residual water and crystal water in MSWI fly ashes [37]. In the range of 200–600 °C, the weight loss is 2.31%, mainly due to the decomposition of $Ca(OH)_2$ and the emission of $CO_2$ in the closed cavity of the ashes itself [38]. The decreasing rate of 600–750 °C significantly accelerated. The weight loss of 5.13% is caused by the decomposition of carbonate such as $CaCO_3$ [39]. The fourth weightlessness of 6.86% occurs in the range of 750–900 °C. The obvious weight loss peak is mainly due to $SO_2$ escape, chloride ion release, salt decomposition, and evaporation during $CaSO_4$ decomposition [33].

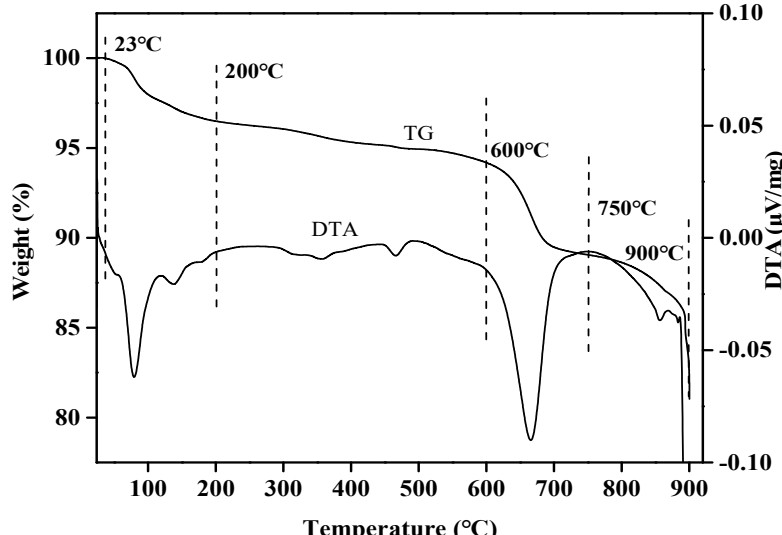

**Figure 2.** Thermogravimetric analysis of MSWI fly ash. (TG: measurements of the change in mass of material with temperature, DTA: the change in measured temperature with program temperature).

### 3.2. Seasonal Variation of Main Alkali Metal Ions and Anions

Table 2 shows the content of major alkali metal ions and anions in MSWI fly ashes measured by GB/T176–2017. Many alkali metal ions are present in the form of volatile chlorides, such as NaCl and KCl. The total content of main alkali metal ions and anions in MSWI fly ashes varies with seasons, being higher in spring and winter, 63.06% and 50.75%, respectively, but lower in autumn and summer, 40.46% and 36.75%, respectively. Additionally, this trend is the same for $K^+$ and $Na^+$, as observed in Table 1. Meanwhile, the alkali metal ions and anions in spring and winter exhibit the same order: $Cl^- > Na^+ > K^+ > Ca^{2+} > SO_4^{2-}$. The same is shown for summer and autumn but the order changes to $Cl^- > Ca^{2+} > K^+ > Na^+ > SO_4^{2-}$. $Cl^-$ ranks first and has the high content because chlorides are volatile and condensed in the ashes during incineration [40]. The augment of $Ca^{2+}$ concentration in MSWI fly ash obtained in summer and autumn stems from the mass utilization of lime slurry for neutralization of acid gas during the incineration of increasing amounts of municipal solid waste [19,41].

**Table 2.** The contents of main alkali metal ions and anions in different seasons (%).

|  | $Cl^-$ | $K^+$ | $Na^+$ | $Ca^{2+}$ | $SO_4^{2-}$ | Total |
|---|---|---|---|---|---|---|
| Spring | 27.64 | 10.69 | 17.45 | 7.28 | 0.44 | 63.06 |
| Summer | 17.18 | 4.69 | 4.59 | 10.29 | 0.38 | 36.75 |
| Autumn | 25.02 | 4.56 | 4.35 | 6.53 | 0.76 | 40.46 |
| Winter | 22.01 | 9.36 | 12.56 | 6.82 | 0.27 | 50.75 |

### 3.3. Seasonal Variation of HMs

The concentrations of HMs in different seasons are shown in Figure 3, and the annual average and standard deviation of each HMs are shown in Table 3. Zn has the highest concentrations among the HMs discussed, followed by Pb, Cu, Cr, Cd, As, and Ni. Zn also fluctuates greatly in the four seasons: 2829.76 mg/kg in spring, 3576.54 mg/kg in summer, 4328.79 mg/kg in autumn, and 3110.76 mg/kg in winter, with the annual average at 3488.56 mg/kg. The contents of Ni (7.74–22.54 mg/kg) and Cd (120.57–149.99 mg/kg) are relatively stable in the four seasons, fluctuating within 30 mg/kg. The concentrations of HMs are not only related to the composition of waste incineration but also closely related to its own melting and boiling points [1]. The boiling points of heavy metal (Zn, Pb, Cd, and Cu) chlorides are below 1000 °C, thus, these metals tend to be volatile and enriched in MSWI fly ashes. The melting point and boiling point of Ni are 1455 °C and 2732 °C, respectively, thus, most of the non-volatile Ni remains in the MSWI bottom ashes [42]. Meanwhile, the contents of these seven HMs are higher than their soil backgrounds, especially for Pb, Zn, and Cd, i.e., the contents of Pb, Zn and Cd in autumn (the highest seasonal value) are 42, 77, and 1260 times higher than the Beijing soil background values, which are 24.60, 57.50, and 0.119 mg/kg, respectively [43,44]. In addition, Pb, Cu, Zn, Cr, and As show a significant seasonal variation. Almost all HMs are higher in summer and autumn and lower in spring and winter except Cr, different with the variation in alkali metal ions and anions, as shown in Table 1. The composition of MSW has a decisive influence on the content of HMs in fly ash [45]. People's diet structure changes with the seasons. In summer and autumn, summer food and fruits are available on the market, resulting in a large number of perishable components such as peels and a high content of HMs in various plastic packaging and packaging paper [5]. The chlorine element in kitchen waste such as peels enhance the volatilization of highly volatile heavy metals, leading to an increase in the concentration of HMs [46]. In addition, in summer and autumn, microorganisms are highly active and their metabolism produces volatile fatty acids, acetic acid, and other corrosive acidic substances. The heavy metal concentration of waste entering the incinerator is high, which is eventually adsorbed into the fly ash [47]. Refer to Supplementary Materials for further relevant information.

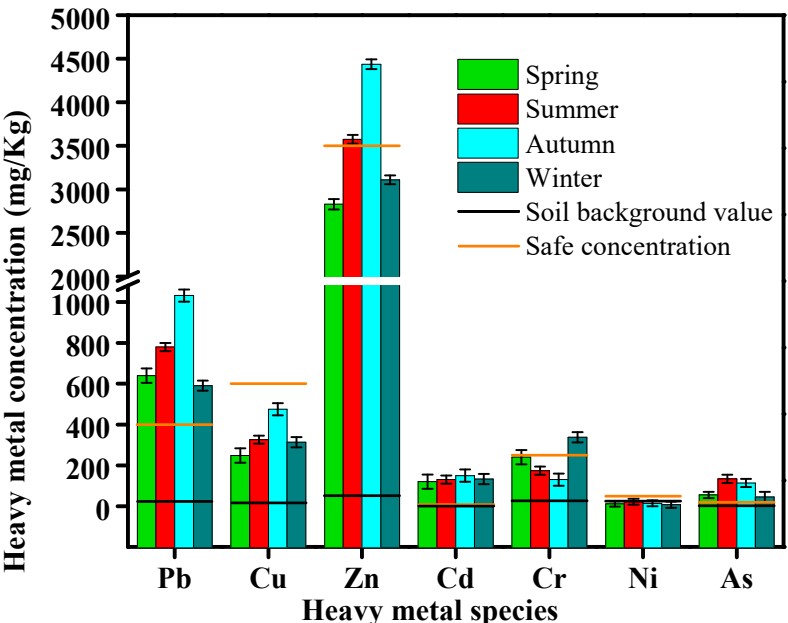

**Figure 3.** Contents of heavy metals in different seasons.

**Table 3.** The average and standard deviation of four seasons (mg/L).

| Element | Pb | Cu | Zn | Cd | Cr | Ni | As |
|---|---|---|---|---|---|---|---|
| Average | 760.16 | 340.84 | 3488.56 | 133.56 | 220.82 | 14.13 | 87.43 |
| Standard deviation | 171.31 | 82.88 | 609.17 | 10.57 | 77.94 | 5.53 | 37.61 |

*3.4. Seasonal Variation of Leaching Toxicity*

HMs leaching toxicity is an important index to evaluate the environmental impact of MSWI fly ashes [13]. Table 4 shows the leaching toxicity of HMs in different seasons using the HJ/T299–2007 and TCLP procedures. The formula for calculating the leaching rate is as follows [10].

$$\text{Leaching ratio} = \frac{\text{leaching concentration}}{\text{Total concentration}} * 100\%, \tag{5}$$

**Table 4.** Leaching characteristics of heavy metals from MSWI fly ash in different seasons (mg/L). (ND means below the detection limit).

| | Zn | Pb | Cu | Cr | Ni | As | Cd |
|---|---|---|---|---|---|---|---|
| GB 5085.3-2007 Standard [7] | 100 | 5 | 100 | 15 | 5 | 5 | 1 |
| Spring | 0.31 | 3.28 | 0.33 | 0.30 | ND | ND | ND |
| Summer | 1.49 | 9.36 | 0.24 | 0.10 | ND | ND | ND |
| Autumn | 0.85 | 10.66 | 0.21 | 0.09 | ND | ND | ND |
| Winter | 0.50 | 5.99 | 0.23 | 0.16 | ND | ND | ND |
| Average | 0.79 | 7.32 | 0.25 | 0.16 | ND | ND | ND |
| Standard deviation | 0.40 | 2.58 | 0.41 | 0.07 | ND | ND | ND |
| Leaching rate | 0.02% | 0.96% | 0.07% | 0.07% | ND | ND | ND |
| TCLP standard | 50 | 5 | 50 | 5 | 10 | 5 | 1 |
| Spring | 0.44 | 3.23 | 0.09 | 0.17 | ND | ND | ND |
| Summer | 1.52 | 6.33 | 0.14 | 0.07 | ND | ND | ND |
| Autumn | 1.08 | 4.98 | 0.09 | 0.08 | ND | ND | ND |
| Winter | 0.51 | 2.73 | 0.07 | 0.10 | ND | ND | ND |
| Average | 0.89 | 4.32 | 0.10 | 0.11 | ND | ND | ND |
| Standard deviation | 0.44 | 1.43 | 0.03 | 0.04 | ND | ND | ND |
| Leaching rate | 0.03% | 0.57% | 0.03% | 0.05% | ND | ND | ND |

According to the method of HJ/T299–2007 [48], the average leaching concentration of Zn in the four seasons is 0.79 mg/L and the leaching rate is as low as 0.02%. The leached fraction of Zn in each season does not exceed the threshold (100 mg/L). The leaching concentration of Pb is 7.32 mg/L, the highest rate among all HMs (0.96%). The leached fractions of Pb in summer, autumn, and winter all exceed the threshold value (5 mg/L). The leaching concentrations of Cu and Cr are 0.25 mg/L and 0.16 mg/L, and the leaching concentrations of both Cu and Cr in the four seasons are below the threshold (Cu is 100 mg/L and Cr is 15 mg/L).

TCLP leaching results show that the average leaching concentration of Zn is 0.89 mg/L, that the leaching rate is 0.03%, and that the leaching content in all four seasons does not exceed the limit threshold (50 mg/L). The average leaching concentration of Pb is 4.32 mg/L, and the leaching rate is 0.57%. Only the Pb leaching content in summer (6.33 mg/L) exceeds the threshold (5 mg/L). The average leaching concentration of Cu and Cr is 0.10 mg/L and 0.11 mg/L, and the leaching concentration of the two HMs is lower than the safety threshold for the four seasons. This result can be confirmed against previous research [27].

A comparison of the two leaching methods shows that the leaching rate of Zn was 0.03% using HJ/T299–2007 and 0.025% using TCLP. Pb, Cu, and Cr all have high leaching concentrations using HJ/T299–2007. Using both methods, the leaching rate of HMs is less than 1% or below the detection limit.

The HMs content and leaching concentration in MSWI fly ash showed that the average content of Zn is the highest at 3488.55 mg/kg and that Zn has the second highest leaching concentration. The average concentration of Pb is 760.15 mg/kg, ranking second, but its leaching concentration is the highest. The order of Cu and Cr contents is consistent with the order of their leaching concentration. This shows that the leaching results of HMs not only correlated with the content of HMs but also are related to the leaching method. A leaching result of less than 1% may be the result of the heavy metals being confined in the lattice and grain boundary [36]. No clear relationship between leaching concentration and heavy metal speciation was found.

Leaching concentrations at different initial pH values were further studied (shown in Table 5). The leaching concentration of HMs in MSWI fly ash reached the highest level at pH = 2 (12.84 mg/L for Zn, 5.49 mg/L for Pb, 0.34 mg/L for Cu, and 0.17 mg/L for Cr), indicating that the leaching behavior of HMs is more active in a strongly acidic environment. With the increase in pH value, the leaching concentration of metals in MSWI fly ashes shows a downward trend (at pH = 2 to 8). However, the leaching concentrations of Pb and Zn increase from 3.15 mg/L to 4.38 mg/L and from 0.59 mg/L to 2.76 mg/L when pH value is between 8 to 12. As Pb and Zn belong to amphoteric metal, which could be redissolved in a strong alkaline solution; part of $Pb(OH)_2$ could be complexed with $OH^-$ to generate $Pb(OH)_3^-$ and $Pb(OH)_4^{2-}$, and Zn could also be complexed with $OH^-$ to generate $Zn(OH)_3^-$ and $Zn(OH)_4^{2-}$. However, the solubility of Zn is weaker than that of Pb [30]. The concentration of Cr also increases slightly when pH increased from 8 to 12. This is because of the formation of the oxyacid anion since its solubility increases under alkali environment [30]. Under different pH values, the leaching concentrations of Cu and Cr are in the range of 0.04–0.34 mg/L and 0.07–0.1 mg/L. Cd, Ni, and As cannot be detected in the investigated pH range.

**Table 5.** Leaching concentrations under different pH (mg/L).

| Metals | Zn | Pb | Cu | Cr | Cd | Ni | As |
|---|---|---|---|---|---|---|---|
| pH = 2 | 12.84 | 5.49 | 0.34 | 0.17 | ND | ND | ND |
| pH = 4 | 8.97 | 5.11 | 0.09 | 0.16 | ND | ND | ND |
| pH = 6 | 3.64 | 3.52 | 0.04 | 0.09 | ND | ND | ND |
| pH = 8 | 0.59 | 3.15 | 0.08 | 0.07 | ND | ND | ND |
| pH = 10 | 2.44 | 4.49 | 0.1 | 0.14 | ND | ND | ND |
| pH = 12 | 2.76 | 4.38 | 0.05 | 0.09 | ND | ND | ND |

*3.5. Seasonal Variation of Chemical Speciation for HMs*

The pollution of MSWI fly ashes is influenced by HMs total concentration and their chemical speciation [10] since its mobility, bioavailability, and leaching toxicity are mainly affected by chemical speciation [49]. The HMs acid-soluble fraction is easily released, which results in potential harm to the environment and humans [10,27]. The HMs reduction fraction is also easily released in a reductive environment, but the possibility of releasing HMs in a reduction fraction is lower than that of the acid-soluble fraction [10,27]. HM oxidation fractions are more stable than the reduction fractions and less likely to be released into the environment [50]. The residual fraction is the most stable fraction, and this part is difficult to release into the environment [27]. Figure 4 shows the chemical speciation analysis of HMs in different seasons. As discussed previously, HMs chemical speciation also shows a seasonal variation. For example, the HMs acid-soluble fraction and reduction fraction are higher in spring and winter and lower in summer and autumn, as shown in Figure 4. The HMs reduction fractions are in the ranges of 22.02–77.7% in spring, of 2.65–30.43% in summer, of 0–28.93% in autumn, and of 0–53.8% in winter. In contrast, for the residual fraction, it is higher in summer and autumn and lower in spring and winter.

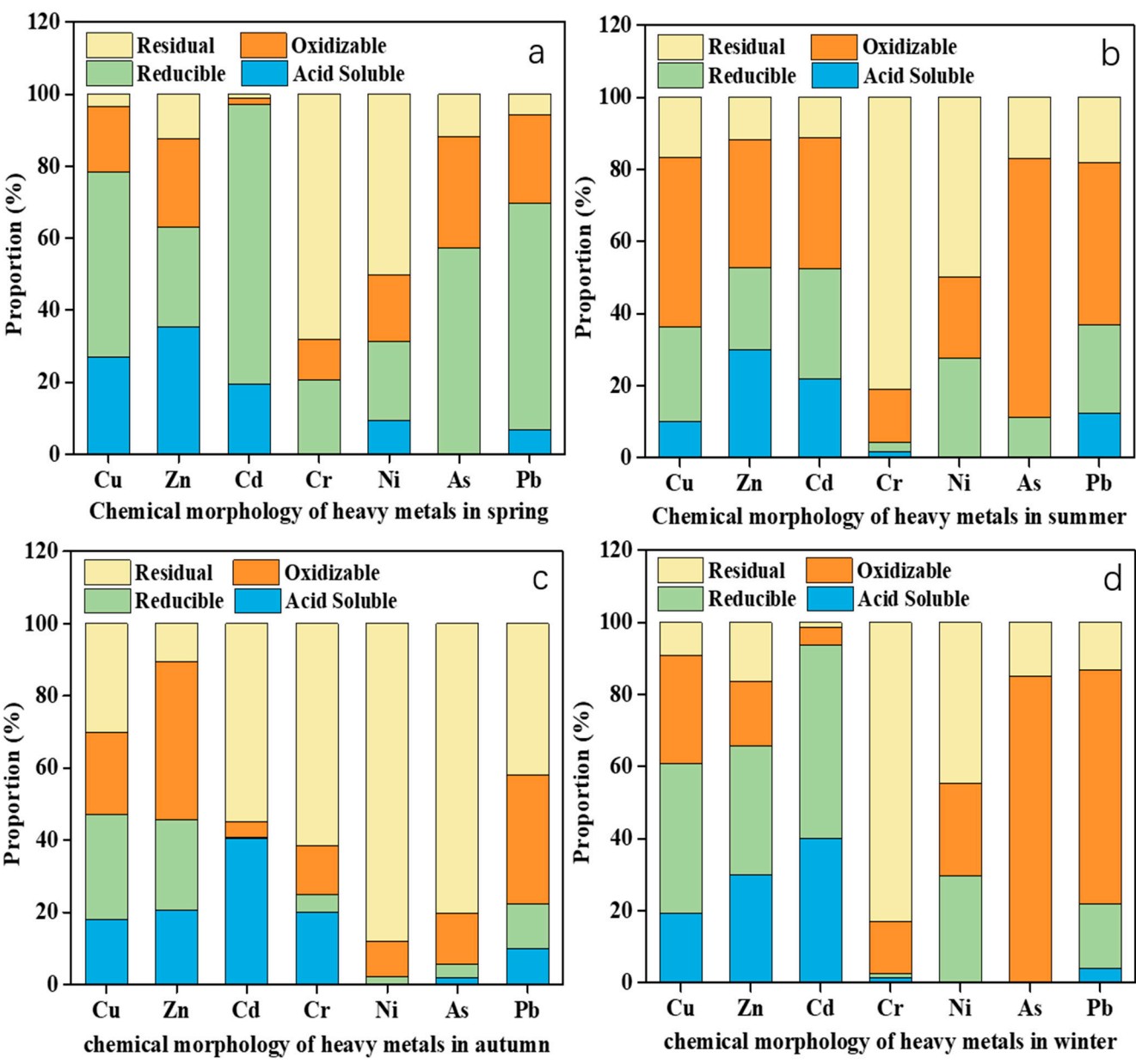

**Figure 4.** Chemical speciation of Beijing MSWI fly ash BCR (**a**–**d**).

Some scholars suggest that, to better understand heavy metal migration, the first three states of BCR should be classified as an active fraction [10]. Based on this hypothesis, Zn, Pb, Cu, and Cd have much higher leaching toxicity than Cr, Ni, and As. Zn can easily be released into the environment since its residual fraction in four seasons is lower than 16.44% and its content is the highest in MSWI fly ashes. Although the Pb acid-soluble fraction in the four seasons is relatively low (3.98–12.49%), the proportion of the active fraction is higher (48.02–87.41%); thus, Pb still shows high mobility. The contents of Cu and Cd in MSWI fly ashes are higher, and the active fraction ratios of Cu and Cd are 69.97–96.59% and 41.59–78.91%, respectively, also representing higher leaching toxicity. The contents of Cr, Ni, and As in MSWI fly ash are lower, and their residual fractions are 68.02–89.49%, 44.71–88.12%, and 11.76–80.16%, which make them less likely to be transferred to the environment.

### 3.6. Seasonal Variation of Bioavailability

As seen in Table 6, the HMs bioavailability of MSWI fly ashes is in the following order: Pb > Zn > Cd > Cu > Cr > As > Ni. As shown in Figure 3 and Table 6, the sequence of HMs total content and bioavailability is not the same. Pb bioavailability ranks first, while its content is second. Zn is on the opposite, it is first in content and fourth in bioavailability, about 24.92 mg/kg. The Zn content is 15.80 times higher than that for Cd, while Zn bioavailability is just twice of that of Cd. This is consistent with previous studies that showed that total concentration is not the key to understanding their bioavailability in the ecosystem [51,52]. Similarly, the bioavailability of MSWI fly ashes shows a strong relation with season: the bioavailability concentrations of Pb, Zn, and Cd in spring and autumn are higher than that in summer and winter, while the concentration of Cu in summer and autumn is higher than that in spring and winter. According to the proportion of acid soluble and reducible fraction, the Pb bioavailability is higher in spring and summer while the Zn, Cd, and Cu bioavailability are higher in spring and winter, which is not completely consistent with the seasonality of bioavailability. These results indicate that bioavailability is determined not only by acid soluble and reducible fraction but also by other chemical forms. Previous studies have shown similar results [53]. Cr, As, and Ni show lower biohazardous risk since the bioavailability concentrations of Cr and As in the four seasons are less than 1 mg/kg [52] and Ni could not be detected. Moreover, we provide some information in the Supplementary Materials on the seasonal variation for the different variables measured in this study in relation to the raw material.

**Table 6.** Bioavailability concentrations of heavy metals in samples (mg/kg).

|         | Pb    | Zn    | Cd    | Cu    | Cr   | As   | Ni  |
|---------|-------|-------|-------|-------|------|------|-----|
| Spring  | 52.61 | 34.12 | 14.29 | 2.67  | 0.80 | ND   | ND  |
| Summer  | 38.84 | 19.92 | 9.76  | 5.86  | 0.37 | 0.53 | ND  |
| Autumn  | 71.47 | 34.92 | 13.11 | 16.18 | 0.43 | 0.67 | ND  |
| Winter  | 34.35 | 10.72 | 9.49  | 1.10  | 1.00 | ND   | ND  |
| Average | 49.32 | 24.92 | 11.66 | 6.45  | 0.65 | 0.30 | ND  |
| Order   | 1     | 2     | 3     | 4     | 5    | 6    | ND  |

## 4. Conclusions

In this paper, the metal mobility and toxicity of MSWI fly ash in different seasons in the Beijing area were analyzed. XRD indicates that the main crystal phases of MSWI fly ashes are $CaSO_4$, $CaCO_3$, and KCl. The total contents of alkali metal ions and anionic ions in the four seasons are 63.06%, 36.75%, 40.46%, and 50.75%. The total contents of Pb, Cu, Zn, and As in MSWI fly ashes in summer and autumn are significantly higher than those in spring and winter, and the contents of Pb, Zn, and Cd in autumn are 42, 77, and 1260 times higher than that in Beijing soil backgrounds, respectively. Under the standard method of GB 50085.3–2007, the leaching concentration of Pb in summer, autumn, and winter exceeded 5 mg/L. Under the TCLP method, only the Pb leaching concentration

exceeded a safe value in summer. The acid soluble and reducing components of HMs are higher in spring and winter. Bioavailability experiments indicate that the bioavailability concentrations of Pb, Zn, and Cd are higher in spring and autumn than in summer and winter. The results show that the MSWI fly ash is a type of hazardous waste. The mobility and toxicity of heavy metals change with seasons. The leaching toxicity of Pb, Zn, and Cd is high, and they easily migrate. This research can provide useful data for the treatment and reuse of MSWI fly ash.

**Supplementary Materials:** The following are available online at https://www.mdpi.com/article/10.3390/su13126532/s1, Figure S1: Correlation of standard curve. Figure S2: Correlation of standard curve. Figure S3: Correlation of standard curve. Figure S4: Correlation of standard curve. Table S1: Recovery rate of the addition of the standard solution (mg/L). Table S2: Test concentration and relative deviation (mg/L). Table S3: Recovery rate of the addition of the standard solution (mg/L). Table S4: Details for HM concentration using the HJ/T299-2007 method (mg/L). Table S5: Details of HM concentration using the TCLP method (mg/L). Table S6: Recovery rate of the addition of the standard solution (mg/L). Table S7: Test concentration and relative deviation (mg/L). Table S8: MSW physical composition of the Haidian and Dongcheng districts (%). Table S9: MSW physical composition of the Xicheng and Shijingshan districts (%).

**Author Contributions:** This work was conducted by various authors as follows: H.Z.: methodology, investigation, and writing—original draft preparation; M.C. and J.S.: methodology, writing—original draft preparation, and investigation; R.W. (Rong Wang), Y.T. and R.W. (Rui Wang): conceptualization, writing—review and editing, and methodology; X.Z., Z.W. and F.Y.: methodology and formal analysis; M.C.: project administration, supervision, and funding acquisition. All authors have read and agreed to the published version of the manuscript.

**Funding:** This work was supported by National Natural Science Foundation of China (21806132); by the National Key R&D Program of China (2018YFC1903500); by the Science and Technology Program of Guizhou Province, China ([2019]2863); by the Sichuan Province Science and Technology plan—Key Research and Development (19ZDYF0911); and by the State Key Laboratory of Solid Waste Reuse for Building Materials' (SWR-2017-004).

**Institutional Review Board Statement:** The study did not involve humans or animals.

**Informed Consent Statement:** Informed consent was obtained from all subjects involved in the study. Written informed consent has been obtained from the patient(s) to publish this paper.

**Data Availability Statement:** The study did not report any data.

**Acknowledgments:** Authors appreciate the constructive suggestions from reviewers and editors that helped improve this paper.

**Conflicts of Interest:** All authors declare no conflicts of interest.

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
