# Peer review of "Seasonal Variation of the Mobility and Toxicity of Metals in Beijing’s Municipal Solid Waste Incineration Fly Ash"

_sustainability, doi:10.3390/su13126532_

Round 1
Reviewer 1 Report
This study investigated the potential toxicity of heavy metal in fly ashes samples collected in a municipal solid waste incineration during different seasons. The authors applied different extraction methods to estimate the soluble fractions of toxic metals. The type of work is interesting to both scientists and policy makers. However, many substantial parts are missing regarding the methodology and discussed results. Full experimental details are lacking for collection of samples, validations of the extraction procedures and chemical analysis. Some assertions are not consistent with the illustrations (figures) and clear connections are missing between results from different section. Many words are missing in the text and some parts were not well written, the English should be reviewed by the native speaker to make the dissertation understandable.
At this state, I cannot support the manuscript for publication in Sustainability. Many items outlined below should be addressed before reconsidering a re-submission.
Introduction
Lines 28-33: this paragraph should include some information on the different management options of MSWI fly ash with some details on the problem connected to the treatment (landfilled or disposed) of these residues. This will help to understand the sentence in line 32-33. Also, information regarding the regulation for treatment of the MSWI fly ash should be considered.
Line 32: add “s” to metal.
Lines 36-38: should be rephrased more clearly. What do you mean by these (line 36)? Please replace “harm of HMs” by “harmful effects of HMs”.
Lines 42-44: the causes (treatments, conditions …) of pH variation should be given.
Lines 44-45: What (parameters) influence the seasonality in the toxicity and mobility of HMs from MSW? Is it connected to the raw material? The authors should develop on this point, especially in the section 3. (Results).
Lines 50-52: there are not details in the section 2 regarding the analysis of physicochemical properties of MSWI fly ash by X–ray fluorescence spectrum (XRF), X–ray diffraction (XRD) and scanning electron microscope (SEM).
Lines 54-54: references are needed for the different methods.
- Materials and Methods
Major details and are missing in the different subsections. There is not information regarding the QC/QA methods (measurement of certified reference materials and blanks? how many replicates for a sample?). Also as mentioned above, detail should be given for XRF, XRD and SEM as for the other analytical methods.
2.1. Materials
The waste incineration power plant should be presented with details on the treatment capacity, the process and type of incinerated MSW for different seasons.
The sample collection, preparation and treatment of the MSWI fly ash should include more details. How many samples per month and per day? Are the samples representatives of the MSWI fly ash emitted from the plan during a day / month? How do you get samples per month?
Lines 66-68: should be rephrased.
2.2. Element analysis
QC/QA methods are missing.
Line 78: the digestion temperature should be given. How the method was validated?
2.3. Leaching toxicity
The whole section needs to be written and restructured more properly. Many words are missing making the sentences hard to read and to understand for non-expert. Reference are missing for the different methods and there is no QC/QA. The authors should briefly describe the methods instead of referring only to the SI.
Line 89: add “to” after “according”.
Lines 88-90: should be rewritten.
Lines 100-101: How the pH value was adjusted to 2, 4, 6, 8, 10 and 12.
2.4. Chemical speciation
As mentioned above, the authors should briefly describe the methods instead of referring only to the SI.
2.5. Metal bioavailability
See my previous comments. References and QC/QA are missing. The authors should also give details about this EDTA solution and what biological/environment medium it mimics.
- Results 112
3.1. Characteristics of MSWI fly ashes
Line 114: should start by “the”. What do you mean by “tested”? how?
Lines 115-116: “39.18 wt.%, 18.38 wt.%, 18.38 wt.%, 8.97 wt.% and 5.99 wt.%”of what?
Line 117: “Al2O3(3.81 wt.%)” an space is missing.
Lines 117-133: as suggested above, a brief description of the MSWI process in section 2 would help to better understand the different steps influencing the Characteristics of MSWI fly ashes.
Line 136: please replace “scattered” by a more appropriate word.
Lines 138-139: to be rephrased.
Line 140-149: there is no information on how the weight loss was determined. It should be included here or in section 2. A caption should be added in Fig 2 giving the meaning of AG and DTA.
Line 141: please replace “there is” by “it is characterized by”.
3.2. Seasonal variation of main alkali metal ions and anions
Line 154: do you consider the elements in the table as HMs? Please rewrite this sentence.
Line 157: this trend is not valid for Cl-.
Lines 158-159: how can you explain this seasonal change? Should be discussed.
3.3. Seasonal variation of HMs
Lines 167-181: the results discussed here are not consistent with the results in Fig 3. This part should be written to fit with the results in the figure. Some trends are not visible in the figure, a table with mean and standard deviation would be better.
The assertion starting by “The contents of As …”( Lines 167-168) seems not consistent with Fig 3 where concentration of As seems higher than that of Ni.
Lines 170-174: as previously mentioned, a description of the MSWI process with the temperature profile should be definitely given to better understand this part (see my previous comment on this).
Line 174: “…The contents of As …”it’s not valid for Ni.
Lines 175-176: “… the content of Pb, Zn and Cd in autumn (the highest seasonal value) ...”. the authors should be coherent with the table and the previous assertion where they considered that the contents of As are stable whereas this element fluctuate more than Cd during the seasons. Further (lines 177-179) it’s asserted that there is a significant seasonal variation with higher levels in summer and autumn …”. There is no clear seasonal variation observed for Cd and Ni.
Lines 180-181: is it the reason/cause of the seasonal variation? As previously suggested, the authors should discuss this point with more details.
3.4. Seasonal variation of leaching toxicity
Line 187-188: should be rephrased more concisely.
Table 2: should mention if these values correspond to the mean. I suggested adding the standard deviation. What does ND mean in the table? Should be added in caption.
Line 192: Please replace “content” by “leached fraction”. Same applies to line 194.
Line 193: “threshold” or “Standard” in table: would be welcome to define this limit and explain what does exceeding this value imply, especially in the introduction.
Line 201: “Only the leaching content …”for what element?
Line 205- 206: is it really higher for TCLP (0.02 and 0.03%)? please replace “program” by method or protocol.
Line 208-209: please replace rewrite this sentence.
Lines 212-217: should written more clearly. The leaching concentration for both method are very low or insignificant or even negligible. Why? Further, the authors should also discuss the role of the HMs' speciation on the leaching concentration instead of explaining it only by the used method.
Line 218: “(As ..)” seems like HM arsenic.
3.5. Seasonal variation of chemical speciation for HMs
Why the HM with high active fraction show very low leaching toxicity? What does this imply?
3.6. Seasonal variation of bioavailability
The bioavailability should also be given in % and discussed with respect to the chemical speciation, leaching rate and seasonal variation.
Is the seasonal variation for the different variables measured in this study connected to the raw material?
4. Conclusions
The conclusion is not so obvious based on the results and illustrations presented here..
Supplementary Materials
This section is very confusing. It should be written and restructured more clearly with some tables showing the different steps. The authors should avoid starting sentence by “Added …” .
References
Please check the reference list. i.g. journal name is missing for Ref. 19.
Reviewer 2 Report
1. Well presented experimental research. Authors have correctly described the research problem and methodology.
2. Authors presented the results of the study on metal mobility and toxicity of municipal solid waste incineration fly ashes from different seasons in Beijing. They have discussed some leaching toxicity procedures, sequential extraction procedures as well as bioavailability procedures.
3. The presented results proof that the leaching of Pb, Zn and Cd from municipal solid waste incineration ash is highly toxic, easy to migrate and has a great biological hazard. All results indicate that MSWI fly ashes metal mobility and toxicity is varied as the seasons.
4. The manuscript is original, however, Authors should focused their conclusions on the sources of MSWI fly ashes and the variability of the composition of hazardous substances in ashes. Seasonal changes in the composition of heavy metals and environmentally hazardous compounds probably stem from the industrial activity of the inhabitants of the Beijing region. Linking the sources of pollution with the obtained results would significantly increase the scientific value of manuscript.
5. Used instruments met the research standards.
6. The manuscript is well written. The description of the research methodology is clear and understandable.
7. Knowing the sources of pollution, a valuable supplement to the article would be to indicate practical solutions to reduce pollution in MSWI fly ashes.
Reviewer 3 Report
Heavy metals and metaloids are major toxicants and a major health hazard. As authors have pointed out, excessive amounts of these elements in the soil, and especially in water sources, may have severe and lasting effects on local populations.
The authors present data related to migration and leaching toxicity of heavy metals in fly ash that results from incineration of municipal solid waste in Beijing.
The methods used for the analysis are appropriate and the results are significant.
Line 17: what is the allowable/safe limit of Pb? I recommend replacing “the limited threshold” with the exact value.
Line 53: could you cite documents containing descriptions on the TCLP?
Line 121: Fly ash may be used to stabilize CO2 foams in the context of carbon utilization and storage in the subsurface. Here is an example:
Stabilization of CO 2 foam using by-product fly ash and recyclable iron oxide nanoparticles to improve carbon utilization in EOR processes. Sustainable Energy & Fuels 1 (4), 814-822, 2017.
Were the measurements, e.g., those in Tables 1 and 4, done only one time? If there were multiple measurements, have you calculated the errors of measurement and would you be able to report these as well?
Line 173: melting point of Ni: 1455C? boiling point 2732C? please provide references for all such data.
Line 176: “…is 42, 77 and 1260 times higher than the Beijing soil …”. Did you measure Beijing’s soil background values? If no, where is the data coming from? How does the data relate to the incineration/disposal site? If measurements were done as part of this study, please describe those as well, e.g., sites, times of year, procedure, etc. Is the soil background values affected by seasons and precipitation? Finally, I have the impression that fly ash samples were collected over one year. It would be good to clarify this in the manuscript and discuss the potential for variability across the years as well.
Fig. 3: It might be helpful to the readers if the max allowable/safe concentration for each element were added on this graph.
Line 191-197: HJ/T299–200 – please provide references.
Table 2: GB5085.3–2007 Standard – references?
Lines 273 – 275: “… the bioavailability concentrations of Pb, Zn and Cd in spring and autumn is higher than that in summer and winter, while Cu concentration in summer and autumn is higher than that in spring and winter.” I suggest either adding a discussion here to expand on this, or pointing to references for those who are interested. If neither is possible, you may add a short discussion that this is an open question, etc. I wonder if there is more to this than a change in people’s diet.
Round 2
Reviewer 1 Report
The authors attempted to bring some improvement to the manuscript but not enough for me to warrant its publication in Sustainability. The manuscript definitely needs substantial grammatical revisions in. The responses are sometime inconsistent with the part which the authors referred to. Some information which they refer in the answers cannot be found in the revised manuscript. Additionally, some information are only transferred from one part (e.g. sup mat) to another without any significant improvement. Consequently, inappropriate part (Author Contributions) was is cut off from and pasted in the manuscript (section 2.2). Finally, connection is lacking between the key results of the study e.g. bioavailability and the chemical speciation.
Among others items to improved are shown below (in red) along with corresponding responses from the authors.
Response
Thanks for the suggestion. We have rewritten the sentence to Lines 36-38.
With the development of the study on the toxicity of HMs of MSWI fly ashes, the evaluation of harmful effects of HMs were assessed by the total concentration, mobility and leaching toxicity instead of total concentration gradually. This is of great significance for the evaluation and control of HMs pollution.
Lines 40-41: it’s not clear the difference between “total concentration” and “total concentration gradually”.
Response
Thank you for your suggestion. The following sentences have been added in section 2.6.
The composition of the sample was analyzed using XRF (XRF, UltimaIV, Rigaku, Tokyo, Japan). The mineral phase of the samples were analyzed using an XRD (XRD, Axios MAX, PANalytical B.V., Almelo, Netherlands). The size and morphology of the samples were studied by SEM (SEM, FESEM, S-3400N, HITACHI, Tokyo, Japan). Thermogravimetric analysis (TG) was performed on a TA Q500 apparatus between 20 and 1000°C with a heating rate of 10 K/min under N2 atmosphere.
Section 2.6: This section should be moved after Section 2.1. Details are still missing for the different methods. At least references should be added for all methods, especially for the Thermogravimetric analysis.
Response
This information has been added, Page2, Line 68-70 as follows:
Samples were collected from Lujiashan waste incineration power plant in the southwest of Beijing. The annual domestic waste treatment capacity is 750 t/d and the mechanical grate process is adopted.
In the revised manuscript, this information is added in brackets. Why? Should be a new sentences as written in the response to reviewer and reference should be added.
Response
This information has been added, Page2 Line 71-74 as follows:
Fly ash samples were collected 12 months a year, and 5 kg of fly ash was collected on a fixed day every month. Fly ash samples are collected at the ash outlet of the flue gas purification device in the waste incineration plant.
In the Line 71-74 of the revised manuscript, the following details which the authors refer to are missing “Fly ash samples were collected 12 months a year, and 5 kg of fly ash was collected on a fixed day every month. Fly ash samples are collected at the ash outlet of the flue gas purification device in the waste incineration plant.”
Response
Thanks for the suggestion. Line 66-68 has been revised as following sentence.
Remove larger particles with a 100-mesh sieve, grind and put sample into a self-sealing bag.
Lines 74-75: This sentence is not clear. It needs grammatical correction.
Response
Line78 has been revised to “Then, put them in Multiwave Pro and digestion at 180 degrees (Anton Paar, Graz, Austria).”
In the process of measuring data, the accuracy of experimental data is guaranteed from the following aspects.
The digested efficiency is not discussed. The information (QC/QA) given bellow regarding the sampling analysis by ICP-OES is very important to give more credit to the work and should be added in supplementary material and reference in the text. Unit should be added in the table.
The information from lines 89-127 are moved from previous sup mat without significant grammatical editing improvement. Same comments apply to others sections.
The section “Author Contributions” is inappropriate here (section 2.2).
Response
Thanks for your suggestion. The whole section has been revised to the following sentence.
(1) Leaching toxicity by the sulfuric acid & nitric acid method
According to "Solid Waste–Extraction procedure for leaching toxicity–Sulfuric acid & nitric acid method" (HJ/T299–2007) [1], adding 100.00 g sample to the extractant and make sure the mixture ratio of solid sample and liquid extractant (S/L) was 1:10. The extractant was prepared as following: Using mixed acid (the mass ratio of sulfuric acid to nitric acid is 2:1) to adjust pH value as 3.20±0.05, Then the mixture was oscillated for 18±2 h (30±2 r/min). After that, filtering the mixture by a 0.45 μm microporous membrane. Finally, measuring the HMs in the samples by ICP–OES.
As for most part of the manuscript, this section is not well written and punctuation is not respected. Same comment applied to the following section where the authors moved some parts from previous sup mat without improvement. The authors should consider using correct sentence instead of enumerating or listing successively some actions. The details regarding the QC/QA should be added in sup mat and referenced in the text.
(2) Toxic leaching of solid waste–TCLP [2]
75.00 g sample and extractant (Adding 5.00 g sample to 96.5 mL deionized water and measuring the pH, selecting the extractant 1 at pH<5.00, and select the extractant 2 at pH>5.00. (Extractant 1: 5.7 mLCH3COOH was mixed with 500 mL deionized water, and 64.3 mL NaOH was added for constant volume to 1L. The pH value was adjusted to 4.93±0.05 by HNO3 or NaOH. Extractant 2: put 1.5 L ultra–pure water into a 2 L beaker and add CH3COOH to make the pH value 2.88±0.05)) were mixed with the S/L is 1:20, then oscillated for18±2 h (30±2 r/min) at room temperature and filtered by a 0.45 μm microporous membrane. using ICP–OES to measure the solution.
See my previous comment.
Response
Thank you for your suggestion. line 88-90 has been changed as following sentence.
Using mixed acid (the mass ratio of sulfuric acid to nitric acid is 2:1) to adjust pH value as 3.20±0.05.
See my previous comment. Should be written using correct sentences.
Response
Measure the pH of the liquid in the beaker with an acidimeter, and add the pH adjusting solution with a burette to keep the pH of the liquid in the beaker at 2, 4, 6, 8, 10 and 12. pH adjusting solution :1 part volume of nitric acid is mixed with 99 parts volume of water or 1 part volume of sodium hydroxide is mixed with 99 parts volume of water.
See my previous comment.
Response
It has been revised.
See my previous comment.
Response
EDTA-2Na solution purchased from Sinopharm Chemical Reagent Co. Ltd. EDTA-2Na solution was used to simulate the chelation and adsorption of HMS in MSWI fly ash by plant root or plant cell organic acid exudates [3].
See my previous comment.
Response
It has been revised.
Same question: “39.18 wt.%, 18.38 wt.%, 18.38 wt.%, 8.97 wt.% and 5.99 wt.%”of what?
This table 2 is added in the revised manuscript and is not referenced anywhere in the text.
Response
It has been revised according to the suggestion. The contents of As (7.74–22.54 mg/kg) and Ni (45.67–134.16 mg/kg) are relatively stable in the four seasons, fluctuating within 150 mg/k was modified as the contents of Ni (7.74–22.54 mg/kg) and Cd (120.57–149.99 mg/kg) are relatively stable in the four seasons, fluctuating within 30 mg/kg.
Is there a safe concentration for Cd?
Response
Thank you for your suggestion. According to the references [16-18], the unit of bioavailability is mg /Kg
Figure 3 show that the total concentrations of Pb, Cu, Zn, Cd and As in summer and autumn are higher than those in spring and winter. Table 5 shows the bioavailability of Pb, Zn and Cd are higher in spring and autumn, while Cu and As are higher in summer and autumn. It is indicated that bioavailability was not a single linear with total content. This result has been reported in the previous research [4]. And other studies have pointed out that bioavailability is directly related to acid soluble and oxidizable forms, and has potential relationship with other chemical forms [52].
As previously mentioned, in this study the authors should discuss if there is connection between their results from the bioavailability and the chemical speciation instead of citing only the litterature. Line 373-374: reference is missing for this assertion (“Cr, …. less than 1 mg/kg, …”).
Response
Research pointed out that the main sources of heavy metals in MSW are kitchen waste, ash, plastic, paper [11, 19]. The composition changes of domestic waste in Beijing are shown in the table below.
Table 9 MSW physical composition of Haidian and Dongcheng districts (%) [20]
Interesting information to be added in sup mat and referenced in the relevant place in the text.
